

# Remimazolam *versus* propofol for procedural sedation: a meta-analysis of randomized controlled trials

Yu Chang[1,*], Yun-Ting Huang[2,*], Kuan-Yu Chi[3,4] and Yen-Ta Huang[1]

[1] Department of Surgery, National Cheng Kung University Hospital, National Cheng Kung University, Tainan, Taiwan

[2] Department of Anesthesiology, National Cheng Kung University Hospital, College of Medicine, National Cheng Kung University, Tainan, Taiwan

[3] Department of Internal Medicine, Taipei Medical University Hospital, Taipei, Taiwan

[4] Department of Education, Center for Evidence-Based Medicine, Taipei Medical University Hospital, Taipei, Taiwan

[*] These authors contributed equally to this work.

Corresponding author
Yen-Ta Huang,
uncleda.huang@gmail.com

## ABSTRACT

**Background**. To improve patient tolerability and satisfaction as well as minimize complications, procedural sedation has been widely used. Propofol is the most widely used agent for induction of anesthesia and sedation by anesthesiologists. With a different mechanism compared to propofol, remimazolam is a new short-acting GABA-A receptor agonist. It is an ester-based benzodiazepine. This meta-analysis aims to clarify the efficacy and safety of remimazolam versus propofol for procedure sedation.

**Methods**. Electronic databases were searched for randomized controlled trials (RCTs) comparing efficacy or safety of remimazolam versus propofol. Meta-analysis were conducted using RStudio with "metafor" package with random-effects model.

**Results**. A total of twelve RCTs were included in the meta-analysis. The pooled results demonstrated that patients with remimazolam for procedural sedation had lower risk of bradycardia (OR 0.28, 95% CI [0.14–0.57]), hypotension (OR 0.26, 95% CI [0.22–0.32]), and respiratory depression (OR 0.22, 95% CI [0.14–0.36]). There was no difference in the risk of developing postoperative nausea and vomiting (PONV) (OR 0.65, 95% CI [0.15–2.79]) and dizziness (OR 0.93, 95% CI [0.53–1.61]) between the remimazolam and propofol groups. Using remimazolam for procedural sedation is significantly associated with less injection pain compared to propofol (OR 0.06, 95% CI [0.03–0.13]). Regarding the sedation efficacy, there was no difference in sedation success rate or time to loss of consciousness, recover and discharge between the remimazolam and the propofol groups.

**Conclusions**. Based on our meta-analysis, patients receiving procedural sedation with remimazolam had lower risk of bradycardia, hypotension, respiratory depression and injection pain compared with propofol. On the other hand, there was no difference in sedation success rate, risk of PONV, dizziness, time to LOC, recovery and discharge between these two sedatives.

**PROSPERO registration number**. CRD42022362950

## INTRODUCTION

Advancements in the medical care system in the 21st century have helped to increase accessibility to health care services, including medical procedures (*e.g.*, colonoscopy *Stock, Haug & Brenner, 2010*, gastrointestinal endoscopy *Zagari et al., 2016*, and hysteroscopy *Orlando & Bradley, 2022*) that are now more often performed worldwide. During procedures, patients may feel anxiety, fear, and physical or emotional stress due to discomfort or pain, and such distress can lead to complications or unfavorable outcomes (*Morrison et al., 1998*; *Zubarik et al., 2002*). Procedural sedation has been widely used to improve patient tolerability and satisfaction and subsequently minimize these complications (*Tobias & Leder, 2011*).

Propofol is the most widely used agent for inducing anesthesia and sedation by anesthesiologists (*Fulton & Sorkin, 1995*; *Trapani et al., 2000*). Propofol is a potent intravenous sedative/hypnotic agent with a rapid onset of action and an extremely short half-life, indicating rapid awakening and quick recovery of cognitive functions after anesthesia or sedation (*Fulton & Sorkin, 1995*; *Trapani et al., 2000*). However, several propofol-associated adverse events have been reported in the literature, such as hypotension, bradycardia, respiratory depression, and injection pain (*Marik, 2004*; *Qadeer et al., 2005*; *Jalota et al., 2011*; *Newstead et al., 2013*).

Remimazolam is a new short-acting GABA-A receptor agonist that can address the distinct mechanisms associated with propofol; it is an ester-based benzodiazepine (BZD) and can be rapidly hydrolyzed into inactive metabolites by tissue esterases (*Lee & Shirley, 2021*). The onset of remimazolam also takes approximately 1–3 min and has a short metabolic half-life, thereby providing adequate moderate sedation but faster recovery after intervention (*Lee & Shirley, 2021*). A recent published meta-analysis reported that remimazolam had a lower success rate for sedation or anesthesia and a lower incidence of adverse events than propofol (*Zhang et al., 2022a*). This review by *Zhang et al. (2022a)* included not only patients who underwent procedural sedation but also general surgery and heart surgery, but conceptual heterogeneity was encountered because of the great variations in the included studies. Other emerging studies have reported the better suitability of remimazolam for procedural sedation compared with propofol (*Guo et al., 2022a*; *Guo et al., 2022b*; *Qiu et al., 2022*; *Yao et al., 2022*). In the present study, our aim was to conduct a systematic review and meta-analysis to update and clarify the efficacy and safety of remimazolam against propofol for procedural sedation.

## METHOD

We performed a systematic review and meta-analysis following the Cochrane Handbook for Systematic Reviews of Interventions (*Higgins et al., 2021*) and presented our findings according to the Preferred Reporting Items for Systematic Reviews and Meta-Analyses (PRISMA) guidelines. Our study was registered as CRD42022362950 on PROSPERO.

### Study selection

Yun-Ting Huang and Yu Chang independently searched electronic databases, including PubMed, Embase, and Cochrane Library, from January 1900 to December 2022 to identify

relevant studies. In cases of discrepancies, a consensus was reached with senior reviewers, specifically Yen-Ta Huang Additional details about the search are shown in Method S1.

### Eligibility criteria

The three predefined criteria for evidence selection were as follows: (1) randomized controlled trials (RCTs); (2) studies that involved adult patients older than 18 years who underwent medical procedures with procedural sedation using propofol or remimazolam; and (3) studies that reported at least one comparative outcome of remimazolam and propofol.

### Exclusion criteria

The studies were considered ineligible when any of the following criteria were met: (1) they took the form of review articles, case reports, case series, retrospective data analyses, or nonrandomized prospective studies; (2) data were unavailable or irrelevant for the meta-analysis; (3) the trials compared other sedatives in the control group instead of propofol; and (4) they were duplicate publications.

### Data extraction

Two investigators (Yun-Ting Huang and Yu Chang) independently extracted relevant information from eligible articles, including (1) first author's name; (2) publication year; (3) country; (4) study period; (5) American Society of Anesthesiology (ASA) score; (6) sample size; (7) procedure type; and (8) protocol of procedural sedation. Any discrepancy was addressed by reaching a consensus with senior reviewers (Yen-Ta Huang).

### Quality assessment

Two investigators (Yun-Ting Huang and Yu Chang) independently completed a critical appraisal of the included trials using Cochrane Risk of Bias 2.0 (ROB 2.0) (*Sterne et al., 2019*). Any discrepancy was addressed by consulting a third investigator (Yen-Ta Huang).

### Primary and secondary outcomes

Our main objective was to compare the safety of remimazolam and propofol by analyzing the incidence of adverse events following their administration. Additionally, we assessed the efficacy of these two sedatives by measuring their success rate, time to sedation, and postprocedural recovery as secondary outcomes.

### Statistical analysis

Meta-analyses were conducted using RStudio with the "metafor" package (*Viechtbauer, 2010*) (Method S2). For binary outcomes, we extracted the odds ratios (ORs) from the included studies and pooled them by using a random-effects model with the Mantel–Haenszel method. The pooled OR was presented with 95% confidence intervals (CIs) and $p$ values.

For the continuous outcomes, we extracted the mean and standard deviation (SD) from the included studies and pooled the results by using a random-effects model with the inverse variance method. The pooled results of the continuous outcomes were summarized with weighted mean difference (WMD) and presented with 95% CIs and $p$ values.

Heterogeneity was assessed using I-square, with values of $I^2 < 25\%$, $25\% < I^2 < 50\%$, and $I^2 > 50\%$, indicating low, moderate, and high heterogeneity, respectively (*Higgins, Deeks & Altman, 2003*).

## RESULTS

Our search strategy identified 344 references from the electronic databases, with 20 studies for full-text inspection. Ultimately, we included 12 studies (*Guo et al., 2022a*; *Guo et al., 2022b*; *Qiu et al., 2022*; *Yao et al., 2022*; *Cao et al., 2022*; *Chen et al., 2020*; *Chen et al., 2021*; *Shi et al., 2022*; *Tan et al., 2022*; *Zhang et al., 2022b*; *Zhang, Li & Liu, 2021*; *Lu et al., 2022*) for qualitative and quantitative syntheses (Fig. 1).

### Characteristics of the included studies

Table 1 shows that 12 RCTs involving 2,170 patients who underwent medical procedures with procedural sedation (remimazolam = 1,119; propofol = 1,051) were enrolled between 1987 and 2019. Among the 12 RCTs, five studies included patients who underwent colonoscopy, three included patients who underwent upper gastrointestinal (UGI) endoscopy, two included patients who underwent hysteroscopy, one included patients who underwent endoscopic submucosal dissection, and one included patients who underwent endoscopic variceal ligation.

### Risk of bias assessments

Using ROB 2.0 for quality of assessment of the included studies, we identified 10 of the included trials as having a low risk of bias. We encountered limitations in evaluating the risks of bias in the other two studies because of bias in randomization (Fig. 2).

### Safety and efficacy of remimazolam *versus* propofol

The pooled results demonstrated that patients given remimazolam for procedural sedation had a lower risk of bradycardia (OR 0.28; 95% CI [0.14–0.57]; $I^2 = 0\%$; Fig. S1), hypotension (OR 0.26; 95% CI [0.22–0.32]; $I^2 = 0\%$; Fig. S2), and respiratory depression (OR 0.22; 95% CI [0.14–0.36]; $I^2 = 27\%$; Fig. S3). No differences were observed in the risk of developing postoperative nausea and vomiting (PONV) (OR 0.65; 95% CI [0.15–2.79]; $I^2 = 70\%$; Fig. S4) and dizziness (OR 0.93; 95% CI [0.53–1.61]; $I^2 = 41\%$; Fig. S5) between the remimazolam and propofol groups. The use of remimazolam for procedural sedation was significantly associated with less injection pain than the use of propofol (OR 0.06; 95% CI [0.03–0.13]; $I^2 = 37\%$; Fig. S6). Regarding sedation efficacy, no differences were found in the sedation success rate between the remimazolam and propofol groups (OR 1.40; 95% CI [0.41–4.76]; $I^2 = 41\%$; Fig. S7). The pooled results did not demonstrate significant differences in time to loss of consciousness (LOC) (WMD 12.68; 95% CI [−5.55 to 30.9 s]; $I^2 = 99\%$; Fig. S8), recovery (WMD −84.55; 95% CI [−258.34 to 89.23 s]; $I^2 = 1.0\%$; Fig. S9), and discharge (WMD −2.52; 95% CI [−6.82 to 1.78 min]; $I^2 = 99\%$; Fig. S10) with high statistical heterogeneity. The results of our meta-analysis are summarized in Table 2.

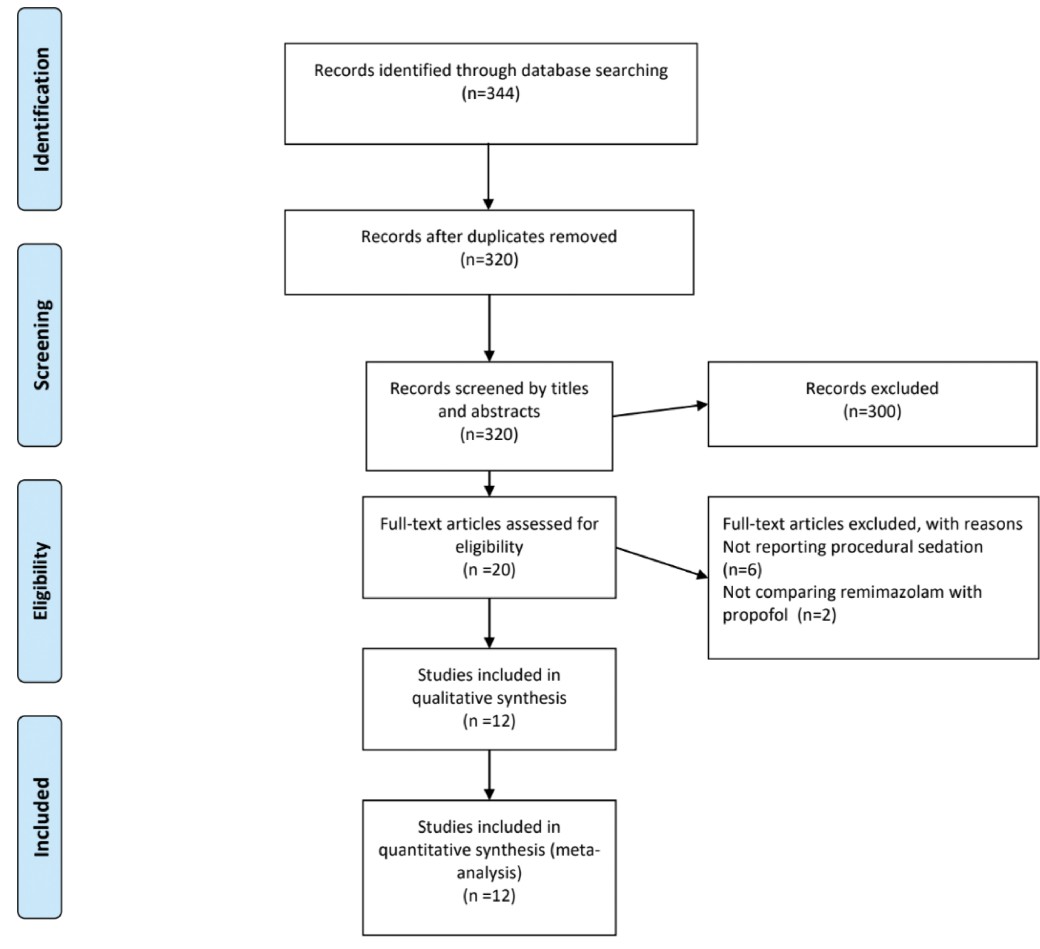

**Figure 1** **PRISMA flowchart diagram.** We initially extracted 344 potential references. Screening the titles and abstracts yielded 20 full-text articles, the eligibility of which was assessed. Eventually, 12 studies were included for qualitative and quantitative syntheses. PRISMA, Preferred Reporting Items for Systematic Reviews and Meta-Analyses.

## DISCUSSION

This meta-analysis focused on patients who received procedural sedation. Our results demonstrated a significantly lower risk of adverse events, including bradycardia, hypotension, respiratory depression, and injection pain, in the remimazolam group than in the propofol group. Regarding sedation efficacy, no differences were observed in sedation success, time to LOC, recovery, and discharge between the two groups.

Procedural sedation may be administered to patients to improve their comfort during diagnostic or therapeutic procedures (*Benzoni & Cascella, 2022*). Remimazolam is a rapidly metabolized and intravenously administered BZD sedative that induces sedation by binding to specific neurotransmitter receptors in the brain. Regarding drug safety, hemodynamic events, including hypotension and bradycardia, are the most common adverse events (*Kampo et al., 2019*). From a drug safety perspective, we compared the adverse events of remimazolam and propofol. Our analysis found that propofol, despite

**Table 1  Characteristics of included studies.**

| Study | Procedures | ASA | Patient number | Intervention |
|---|---|---|---|---|
| Zhang et al. (2022b) | Hysteroscopy | I–II | 90 | Propofol 5 mg/kg/h ($n = 30$) |
| | | | | Remi[t] 0.48 mg/kg/h ($n = 30$) |
| | | | | Remi[t] 0.6 mg/kg/h ($n = 30$) |
| Guo et al. (2022a) | Gastrointestinal endoscopy | I–II | 77 | Propofol 1.5 mg/kg ($n = 38$) |
| | | | | Remi[t] 0.15 mg/kg ($n = 39$) |
| Guo et al. (2022b) | Colonoscopy | I–II | 248 | Propofol 1.5 mg/kg ($n = 124$) |
| | | | | Remi[n] 5 mg/kg ($n = 124$) |
| Shi et al. (2022) | Endoscopic Variceal Ligation | II–III | 76 | Propofol 4–10 mg/kg/h ($n = 38$) |
| | | | | Remi[t] 1–2 mg/kg/h ($n = 38$) |
| Zhang, Li & Liu (2021) | Hysteroscopy | I–II | 82 | Propofol 3–6 mg/kg/h ($n = 41$) |
| | | | | Remi[b] 1 mg/kg/h ($n = 41$) |
| Chen et al. (2020) | Colonoscopy | I–II | 384 | Propofol 1.5 mg/kg ($n = 190$) |
| | | | | Remi[t] 5.0 mg ($n = 194$) |
| Chen et al. (2020) | Upper gastrointestinal endoscopy | I–II | 378 | Propofol 1.5 mg/kg ($n = 189$) |
| | | | | Remi[t] 5.0 mg ($n = 189$) |
| Cao et al. (2022) | Gastroscopy: | II–III | 148 | Propofol: 2 mg/kg ($n = 74$) |
| | | | | Remi[t]: 0.107 mg/kg ($n = 74$) |
| | | | | (Sufentanil 0.15 μg/kg) |
| Qiu et al. (2022) | Endoscopic submucosal dissection | I–III | 56 | Propofol: 2.0 mg/kg ($n = 28$) |
| | | | | Remi[n]: 0.3 mg/kg ($n = 28$) |
| Tan et al. (2022) | Gastrointestinal endoscopy | I–II | 99 | Propofol 1–1.5 mg/kg ($n = 33$) |
| | | | | Remi[t] 0.1 mg/kg ($n = 33$) |
| | | | | Remi[t] 0.2 mg/kg ($n = 33$) |
| Yao et al. (2022) | Colonoscopy | I–II | 132 | Propofol 1 mg/kg ($n = 66$) |
| | | | | Remi[n] 0.2 mg/kg ($n = 66$) |
| Lu et al. (2022) | Upper gastrointestinal endoscopy | I–III | 400 | Propofol 3 g/h ($n = 200$) |
| | | | | Remi[t]: 300 mg/h ($n = 200$) |

**Notes.**

ASA,  American Society of Anesthesiologists classification; remi[b], remimazolam besilate; remi[t], remimazolam tosilate; remi[n], substituent not reported.

being a traditional and widely used sedative, entails a higher risk of varying adverse events. In contrast to the findings of a previous review, our results were reinforced by minimal statistical heterogeneity and a greater sample size with more included studies.

Although pain during injection of sedatives is not considered a major complication, it causes patient discomfort and anxiety. Explanations for pain on propofol injection include phenol-associated irritation of the skin and mucous membrane, vein endothelium, and delayed pain because of the release of mediators such as kininogen from the kinin cascade (Desousa, 2016). Remimazolam is an ester-based BZD with different components, and it theoretically causes less injection pain. Our results support the notion that remimazolam is significantly associated with a much lower risk of injection compared with propofol.

Previous evidence has suggested the antiemetic properties of propofol; in our study, we found a similar incidence of PONV between the remimazolam and propofol groups (Kampo

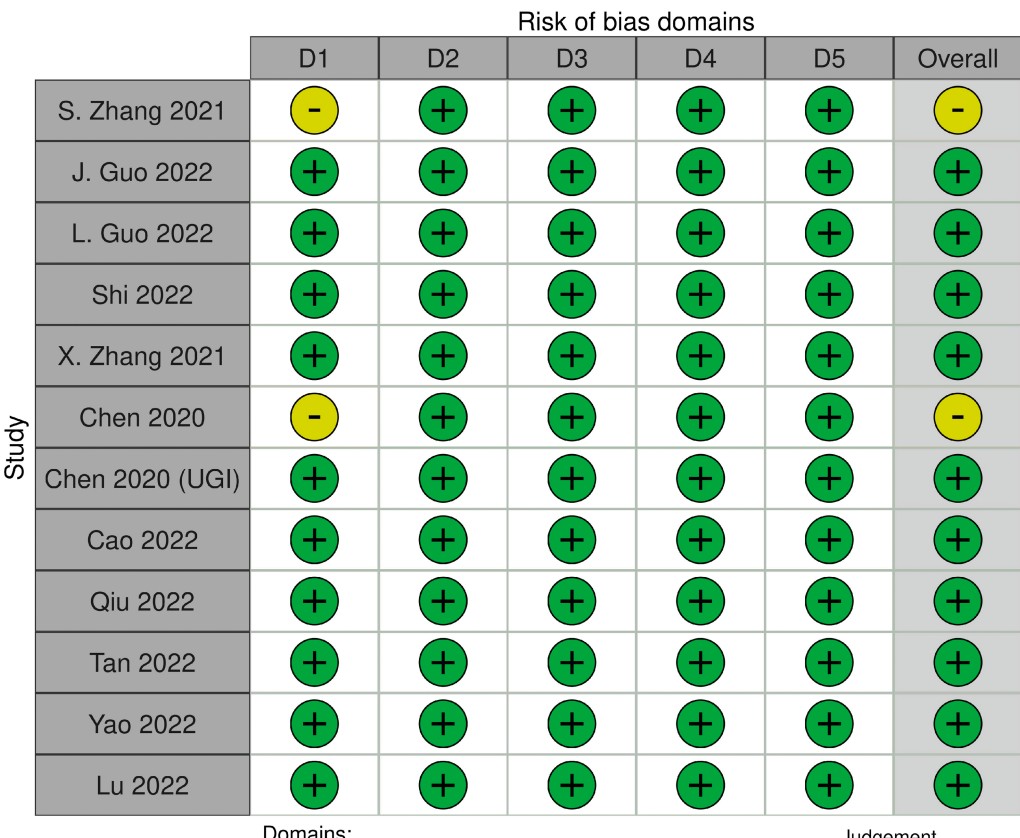

**Figure 2  Visual summary of risk of bias using ROB 2.0 evaluation tool.** Ten included trials are evaluated as low overall risk of bias and two as some concerns for risk of bias.

*et al., 2019*). We believe that the development of PONV or dizziness is multifactorial and associated not only with sedatives themselves but also with the procedure type. For example, *Cao et al. (2022)* studied patients who underwent UGI endoscopy examination and found the incidence of dizziness to be approximately 1.4% in the remimazolam group and 4.1% in the propofol group. *Chen et al. (2020)* studied patients who underwent colonoscopy and determined the incidence of dizziness to be approximately 25% in both the remimazolam and propofol groups.

*Zhang et al. (2022a)* and *Jalota et al. (2011)* reported the lower performance of remimazolam compared with propofol in terms of sedation success rate. However, our meta-analysis did not find significant differences in sedation success rates between the two sedative groups. The discrepancy in results may be explained by the inclusion of various patient populations by the authors, such as those who underwent general anesthesia and procedural sedation, whereas we only included patients who underwent procedural sedation. The comparable time to LOC, recovery, and discharge between remimazolam and

**Table 2** Summary of meta-analyses.

| Outcome | OR | WMD | Number of included studies | $I^2$ |
|---|---|---|---|---|
| Bradycardia | 0.28 (0.14–0.57) | | 6 | 0% |
| Hypotension | 0.26 (0.22–0.32) | | 11 | 0% |
| Respiratory depression | 0.22 (0.14–0.36) | | 11 | 27% |
| PONV | 0.65 (0.15–2.79) | | 8 | 70% |
| Dizziness | 0.93 (0.53–1.61) | | 7 | 41% |
| Injection pain | 0.06 (0.03–0.13) | | 9 | 37% |
| Successful sedation | 1.40 (0.41–4.76) | | 8 | 41% |
| Time to LOC | | 12.68 (−5.55–30.9) | 7 | 99% |
| Time to recovery | | −84.55 (−258.34–89.23) | 9 | 100% |
| Time to discharge | | −2.52 (−6.82–1.78) | 7 | 99% |

**Notes.**

LOC, loss of consciousness; OR, odds ratio; WMD, weighted mean difference.

propofol should be interpreted with caution. First, the pooled results presented extremely high statistical heterogeneity, and the reported mean time to LOC, recovery, or discharge across the studies presented wide variations. Moreover, different procedural durations and complexities likely affected the recovery course and time, a potential bias that could not be justified.

## LIMITATIONS

One of the main limitations of our meta-analysis was related to the variability in study design among the included studies. In particular, the protocols for sedative administration and assessment of adverse events varied across studies. Second, although we included 12 studies, some of them did not report relevant comparative safety or efficacy outcomes, which caused the numbers of studies pooled in the meta-analysis of several outcomes to be less than expected. Third, the high heterogeneity in the overall estimation of time to LOC, recovery, and discharge could not be adequately explained. Future well-designed studies should aim to clarify the reasons for the aforementioned heterogeneity. Fourth, although flumazenil can be used to reverse the effects of remimazolam after a procedure, only two of the studies included in our analysis reported the use of flumazenil. This finding might have affected our comparison of recovery outcomes following procedural sedation between remimazolam and propofol. Finally, all of the included studies were conducted in East Asia; therefore, the races represented in these studies may be primarily Asian. This geographic and racial homogeneity limits the generalizability of our findings to other populations. We hope that in the future, more relevant research will be conducted in other parts of the world to ensure higher applicability of the results to different populations.

## CONCLUSIONS

Our meta-analysis did not show differences in sedation success rate, time to loss of consciousness, recovery, and discharge between remimazolam and propofol. However, in

terms of adverse events, our findings suggest that patients who received procedural sedation with remimazolam had a lower risk of bradycardia, hypotension, respiratory depression, and injection pain than those who received propofol.

## ACKNOWLEDGEMENTS

We are grateful to the English editors of Knowledge Growth Support for helping with language editing.

### Funding
The authors received no funding for this work.

### Competing Interests
The authors declare there are no competing interests.

### Author Contributions

- Yu Chang conceived and designed the experiments, analyzed the data, prepared figures and/or tables, and approved the final draft.
- Yun-Ting Huang conceived and designed the experiments, performed the experiments, prepared figures and/or tables, authored or reviewed drafts of the article, and approved the final draft.
- Kuan-Yu Chi performed the experiments, analyzed the data, prepared figures and/or tables, and approved the final draft.
- Yen-Ta Huang conceived and designed the experiments, authored or reviewed drafts of the article, and approved the final draft.

### Data Availability

The raw measurements are available in the Supplementary File.

### Supplemental Information

Supplemental information for this article can be found online at http://dx.doi.org/10.7717/peerj.15495#supplemental-information.

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
