# Peer review of "Remimazolam versus propofol for procedural sedation: a meta-analysis of randomized controlled trials"

_PeerJ, doi:10.7717/peerj.15495_

## Round 0.1 · original submission · Major Revisions

The reviewers have provided some key comments.

Reviewer 1 ·

Basic reporting

There is no problem with the English language presentation in the main body of the manuscript. The manuscript provides the necessary information for clinicians.

Potential bias may be included in the selection of papers. That is, all papers are from East Asia. All affiliations where studies have been conducted are located in East Asia. The races included in these studies may be primarily Asian. An explanation should be added in the Limitations section.

Experimental design

The study provides clear answers to clinical questions.

An explanation about race should be added in the Limitations section. All referenced papers were scheduled in Asia.

Validity of the findings

The ref#11 cited reports that remimazolam is inferior in points of sedation success rate. A discussion of its uncertainties seems warranted.

The conclusions are concisely stated and there are no scientifically problematic statements.

Additional comments

In the manuscript " Remimazolam versus propofol for procedural sedation: A meta-analysis of randomized controlled trials" by Yu-Chang et.al, submitted to PeerJ for publication, the authors describe the role of remimazolam for procedural sedation. These results provide suggestions for the drug selection for procedural sedation to clinicians.

Reviewer 2 ·

Basic reporting

# I have finished my review process of the submitted article entitled "Remimazolam versus propofol for procedural sedation: A meta-analysis of randomized controlled trials".

・The whole manuscripts was written well. This paper have described clear and easy English to read.
・The authors have described Literature references, sufficient field background and contexts provided.

Experimental design

・Their experimental design is relevant and meaningful.
・Research question is well defined, and PROSPERO registration was confirmed to have been done without any problems.
・Methods section described sufficient detail.

Validity of the findings

The authors showed that remimazolam sedation is safer and had lower risk of bradycardia, hypotension, respiratory depression and injection pain compared with propofol. And there was no diûerence in risk
of PONV, dizziness, time to LOC, recovery and discharge between these two drags.

I think this result have novety and impact about anesthesia field.The whole manuscript is well-written. I have only few minor comments.

・I have a queation about whether remimazolam tosilate or besilate in their paper.Remimazolam has two types of depressants, tosylate and besylate, are they differentiated in the included RCTs? The Authors should describe whether remaimazolam "tosilate" or "besilate" clearly.

・Flumazenil can antagonize the anesthetic effect of remimazolam, which is an excellent point.
So the authors should add the description about whether flumazenil was injected or not.

Reviewer 3 ·

Basic reporting

Summary

Thank you for the invitation to review this important review article.
The authors deserve appreciation for their commitment to conducting a SRMA on remimazolam vs propofol. They produced well-written and relevant work. The objective is well set, the method is well designed, and the results are well described and fairly presented in the discussion section with proper interpretation. However, addressing the following comments may worth giving attention.
Summary

Thank you for the invitation to review this important review article.
The authors deserve appreciation for their commitment to conducting a SRMA on remimazolam vs propofol. They produced well-written and relevant work. The objective is well set, the method is well designed, and the results are well described and fairly presented in the discussion section with proper interpretation. However, addressing the following comments may worth giving attention.
Comments

1. Line 29-33
Comment: The aim of the study is to compare the safety and efficacy profile of Remi vs propofol. In my opinion, providing evidence that show the difference in safety and efficacy may help identifying the gap.
2. Line 29-33, with a completely different mechanism compared to propofol, remimazolamis a new short acting GABA-A receptor agonist.
Comment: Propofol works via facilitating GABAergic action, however, in this paper remimazolam described as having a completely different mechanism. In this same line remimazolam is merged with “is”.
3. Line 34-36
Comment: Mentioning method of analysis, quality and heterogeneity assessment, data bases searched may help readers get information the way this Meta-analysis was done.
4. Keywords are not listed in the abstract
5. Line 77, As the medical care system improved in the 21- century with increased accessibility to
Comment: 21st century
6. Line 79, are performed more and more often worldwide
Comment: more and more often=more often
7. Line 86-87, however, several propofol associated adverse 87 events had been reported in the literature, such as hypotension, bradycardia, and respiratory 88 depression and injection pain
Comment: More citation might be required
8. Primary and secondary outcomes should be stated
9. Line 130, the pooled OR will be presented
Comment: well be??
10. Line 134, continuous outcomes will be summarized
Comment: will be??
11. Line 181-183, Despite propofol, a 182 traditional and widely use sedatives, our meta-analysis showed propofol is associated with higher 183 risk of several adverse event.
Comment: This sentence should be restated
12. Our results would be reinforced by minimal statistical heterogeneity and greater sample size with more included studies than previous review.
Comment: As to my knowledge, I do not understand larger versus smaller number of studies included in meta-analysis, and their effect on the power of the study. Can you, please describe the effect of over-all sample size as taken from all studies and the sample size of each study?
13. Lines 192 and 193, propofol can cause PONV
Comment: Although evidence showed that propofol has antiemetic action, in lines 192, propofol associated PONV risk indicated. Evidence supporting antiemetic action of propofol, among others https://pubmed.ncbi.nlm.nih.gov/31521119/
14. Line 208
Comment: the statement “One of the main limitations of our meta-analysis had to do with the different design of the 209 included studies” requires restatement.
15. Lines 213 and 214
Comment: The sentence “third, the high heterogeneity in the overall estimation of time to LOC, recovery and 214 discharge could not be well-explained and should be clarify in future well-designed study” requires restatement.
16. Conclusion
Comment: conclusion should start with the effect of remimazolam versus propofol on sedation. Then safety issue follows
17. Exclusion criteria: criteria for exclusion of studies should be well-defined in the manuscript

Experimental design

Not applicable

Validity of the findings

The findings of the review are valid.

Additional comments

Summary

Thank you for the invitation to review this important review article.
The authors deserve appreciation for their commitment to conducting a SRMA on remimazolam vs propofol. They produced well-written and relevant work. The objective is well set, the method is well designed, and the results are well described and fairly presented in the discussion section with proper interpretation. However, addressing the following comments may worth giving attention.

---

## Round 0.2 · accepted · Accept

Thanks to the author for the careful revision.

Reviewer 1 ·

Basic reporting

There is no problem with the English language presentation in the manuscript. The manuscript provides helpful information for clinicians.

Potential biases by Asian rase are pointed out in the Limitations section.

Experimental design

The study provides clear answers to clinical questions.

An explanation about race is added in the Limitations section.

Validity of the findings

A discussion of the ref#11 uncertainties are added in the manuscript.

The conclusions are concisely stated and there are no scientifically problematic statements.

Additional comments

This manuscript provides some suggestions for drug selection for procedural sedation to clinicians.

Reviewer 2 ·

Basic reporting

No Comment

Experimental design

No comment

Validity of the findings

The authors have satisfactory responded to all the comments made by the reviewer.

Additional comments

The authors have satisfactory responded to all the comments made by the reviewer.

Reviewer 3 ·

Basic reporting

They prepared a well-written manuscript.

Experimental design

The method is as per the recommendation to conduct systematic reviews and meta-analyses.

Validity of the findings

Authors presented the results maintaining valdity.

Additional comments

The authors addressed all the comments I had provided.